# Molecular Insights into Neurological Regression with a Focus on Rett Syndrome—A Narrative Review

**DOI:** 10.3390/ijms26115361

**Published:** 2025-06-03

**Authors:** Jatinder Singh, Paramala Santosh

**Affiliations:** 1Department of Child and Adolescent Psychiatry, Institute of Psychiatry, Psychology and Neuroscience, King’s College London, London SE5 8AF, UK; paramala.1.santosh@kcl.ac.uk; 2Centre for Interventional Paediatric Psychopharmacology and Rare Diseases (CIPPRD), South London and Maudsley NHS Foundation Trust, London SE5 8AZ, UK; 3Centre for Interventional Paediatric Psychopharmacology (CIPP) Rett Centre, Institute of Psychiatry, Psychology and Neuroscience, King’s College London, London SE5 8AF, UK

**Keywords:** neurological regression, Rett syndrome, epigenome, methylation

## Abstract

Rett syndrome (RTT) is a multisystem neurological disorder. Pathogenic changes in the *MECP2* gene that codes for methyl-CpG-binding protein 2 (MeCP2) in RTT lead to a loss of previously established motor and cognitive skills. Unravelling the mechanisms of neurological regression in RTT is complex, due to multiple components of the neural epigenome being affected. Most evidence has primarily focused on deciphering the complexity of transcriptional machinery at the molecular level. Little attention has been paid to how epigenetic changes across the neural epigenome in RTT lead to neurological regression. In this narrative review, we examine how pathogenic changes in *MECP2* can disrupt the balance of the RTT neural epigenome and lead to neurological regression. Environmental and genetic factors can disturb the balance of the neural epigenome in RTT, modifying the onset of neurological regression. Methylation changes across the RTT neural epigenome and the consequent genotoxic stress cause neurons to regress into a senescent state. These changes influence the brain as it matures and lead to the emergence of specific symptoms at different developmental periods. Future work could focus on epidrugs or epi-editing approaches that may theoretically help to restore the epigenetic imbalance and thereby minimise the impact of genotoxic stress on the RTT neural epigenome.

## 1. Background

Methylation is a crucial aspect of brain development and maturation [1]. It is an essential epigenetic mechanism operating at transcriptional and post-transcriptional levels. Epigenetic mechanisms can include modifications to DNA, i.e., DNA methylation and modifications to RNA and proteins, and these mechanisms operate across lifespans. The best-studied epigenetic mechanism is DNA methylation [2]. Proteins that have a methyl-CpG binding domain (MBD) can ‘read’ methylated DNA and are broadly separated into three families [3], and one of these is the methyl-CpG-binding protein 2 (MeCP2) [4]. The MeCP2 is an important reader of DNA methylation [5], and pathogenic changes in readers lead to aberrant DNA methylation, which has been associated with various neurodevelopment and neurological disorders [6].

### Impaired Methylation and Neurological Disorders

The methylation landscape within the brain is dynamic, and readers, writers and erasers fine-tune DNA methylation. Both genetic and environmental elements can influence these processes. Impairments in epigenetic processes have been associated with various disorders that span across the neurological and neurodevelopmental spectrum [7]. While this narrative review is not an exhaustive list of impaired methylation in all neurological disorders, there are some noteworthy aspects to mention. Defects in de novo methylation caused by a mutation in the gene coding the DNA methyltransferases (*DNMT*) enzyme *DNMT3B* have been implicated in immunodeficiency-centromeric instability facial anomalies (ICF) syndrome (OMIM #242860) [8]. In neurodegenerative disorders such as multiple sclerosis (MS, OMIM #126200), distinct patterns are observed in the methylation of genes involved in inflammation, myelination and the immune response [9]. In Huntington’s disease (HD, OMIM #143100), modifications in DNA methylation have been linked with the expression of mutant huntingtin (HTT) [10], and DNA methylation may also be in part responsible for the tissue-specific HTT transcription [11]. When viewed through the lens of neurodevelopmental disorders, DNA methylation may also affect the regulation of imprinting loci [12]. Prader–Willi syndrome (PWS, OMIM #176270) has been categorised as a genetic and epigenetic disorder caused by defects in genomic imprinting in the 15q11.2-q13 locus [13]. Genomic imprinting is a complex process of differential DNA methylation and gene expression [14]. In PWS, CpG island methylation of the Small nuclear ribonucleoprotein polypeptide N (*SNRPN*) gene has been suggested to be the most highly methylated DNA region in the PWS imprinting centre [6]. Like PWS, Angelman syndrome (AS, OMIM #105830) is another imprinting disorder caused by genetic or epigenetic anomalies within the 15q11.2-q13 locus [15]. Some other disorders, such as Rubinstein–Taybi syndrome (RSTS, OMIM #180849), might not be directly influenced by DNA methylation but have proteins involved in chromatin remodelling that alter DNA methylation patterns [16]. While disrupted DNA methylation is associated with various disorders, a CpG island methylator phenotype has also been identified in brain tumours, facilitating tumour classification [17]. Even though some of these disorders, such as Angelman syndrome, HD, PWS, and RSTS, fall under the rubric of rare disorders, all will have a significant bio-psycho-socio-ecological impact (Figure 1). The management of these disorders is complex, and treatment requires input from several specialists. Patients are not adequately supported, resulting in delays in treatment and consequent deterioration. Families of children with disorders arising from impaired methylation face an immutable burden and often diagnostic isolation.

Previous studies have shown that MeCP2 can repress gene transcription by interacting with histone deacetylase complex [18]. A more recent study in murine brains has shown that MeCP2 is an important component of heterochromatin condensates in cells and may facilitate the separation of heterochromatin and euchromatin [19]. Changes in heterochromatin condensates caused by pathogenic changes in *MECP2* are one of the disease drivers in Rett syndrome (RTT) (OMIM #312750) [19]. MeCP2 is a critical focal point for various molecular processes, and impairments mainly caused by pathogenic variants in the *MECP2* gene give rise to RTT [20], a systemic disorder impacting several different organ systems, that leads to a wide range of autonomic, cardiac, gastrointestinal, neurological, respiratory, sleep, and skeletal problems. However, clinical severity is variable and genotype–phenotype relationships in individuals with RTT may have a limited prognostic value [21,22,23], so the diagnosis remains clinical [24]. Loss (partial/complete) of acquired spoken language and purposeful hand skills, stereotypic hand movements and gait problems are RTT’s core clinical diagnostic criteria [24]. It can be broadly characterised into four stages: (I) symptom onset stage (onset age: 6 months to 1.5 years), (II) neurological regression (onset age: 1–4 years), (III) plateau or pseudostationary stage where symptoms appear to stabilize (onset: 4 to 10 years), and (IV) late-stage motor decline (onset: 10 to 30 years) [24,25]. In RTT, regression duration can be months to a few years [25]. One of the cardinal features of RTT is a regression of skills such as hand use and communication. However, the mechanisms of neurological regression from the gene to the symptom level have not been adequately described in RTT. In this narrative review, we first summarise key information associated with neurological regression in different disorders. We then provide an overview of neurological regression in RTT by (I) describing the impact of *MECP2* on the genomic landscape and (II) describing how adverse events caused by disruption in the *MECP2* gene can imbalance the epigenome, leading to neurological regression in RTT.

## 2. Methods

### 2.1. Search Strategy

The search strategy for the narrative review was performed as previously described [26] and was based on the Preferred Reporting Items for Systematic Reviews and Meta Analyses (PRISMA) guidelines [27]. The narrative literature review flowchart is provided in Figure 2. For this narrative review, the PubMed database was searched using the search terms from December 2024 and restricted to the last 10 years. A follow-up PubMed database search was also performed in May 2025. Boolean operators were used to link search terms, and a truncation symbol (*) were used to facilitate the search.

### 2.2. Search Terms

The following search terms were used:

Neurological regression AND child* disorders AND genetics

Epigenetics AND Rett syndrome

Regression AND Rett syndrome

### 2.3. Secondary Searching

To enrich the primary search, a snowballing approach was used [28] for secondary searching. This strategy allowed further searching of the literature evidence relevant to the subject matter, such as tracing articles on neurological regression in other childhood disorders and further information on the epigenetics of RTT.

### 2.4. Eligibility Criteria

#### 2.4.1. Inclusion Criteria

Articles, including case studies and reviews.

Studies done in humans and animal model.

#### 2.4.2. Exclusion Criteria

Articles not available electronically.

Not available in the English language.

## 3. Key Findings

Following the PubMed search, 948 articles were screened and assessed for eligibility. After removing duplicates, preprints and non-relevant articles, 25 articles remained. Eight articles were reviews on DNA methylation, epigenetics and *MECP2* in RTT [2,3,6,20,29,30,31,32]. Another was a demographic study of RTT in Ireland that showed a higher regression frequency in the Irish RTT population [33]. Other articles examined developmental pathways of regression [34] and other aspects of regression in RTT [35,36,37], including the role of redox imbalances in developmental regression [38] and a review of studies examining behaviour and educational intervention studies in developmental regression [39]. Some other articles explored limitations in NextGen sequencing in RTT [23] or compared developmental encephalopathies from the RTT natural history study [40] or *MECP2*-allelic disorders [41]. Another study in an animal model showed that cortical defects may emerge before behavioural regression in RTT [42] while another assessed transcriptomic signatures [43]. Other aspects of neurological regression were also identified, such as the impact on the caregiver [44], early developmental delay and neurological regression in metachromatic leukodystrophy [45] and genetic findings in other disorders [46,47,48].

### Neurological Regression

While developmental regression can be viewed as a loss of skills, it has a varied aetiology. There is no consensus on how regression should be managed and how it is captured, which has led others to propose that to understand the nature of regression better, cross-disciplinary efforts are needed to disentangle the complex interplay between the biological and environmental factors of regression [39,49]. Regression is seen in a variety of neurodevelopmental disorders such as Autism Spectrum Disorder (ASD), Phelan McDermid syndrome (PHMDS, OMIM #606232), Childhood Disintegrative Disorder (CDD, Heller’s syndrome), RTT, developmental epileptic encephalopathies such as Landau–Kleffner syndrome (LKS; OMIM #245570) and Continuous Slow Wave Sleep disorder (CSWS) but may also be part of other conditions such as Progressive Intellectual and Neurological Deterioration (PIND) or childhood dementia [50]. Neurological regression characterised as Down Syndrome Regression Disorder (DSRD) has also been identified in Down syndrome [51]. Some evidence also points towards regression in childhood-onset schizophrenia [52] and post-traumatic stress disorder [53].

Trajectories of regression vary across disorders. In CDD, the onset of regression was 3 years 2 months (±1 year 1 month) [54], while in PHMDS, one study found that regression occurred in 43% of patients [55] with an onset of 6 years [50]. Regression may also manifest at different stages in PHMDS [56]. A case report has also shown a rapid onset of developmental regression in PHMDS due to a dual diagnosis [46]. In PHMDS, those with class I deletions have more regression in skills [47]. Moreover, while evidence suggests that the onset of regression in CDD emerges gradually between 6–12 months [57], the trajectory of regression in CDD may be quicker than typically seen in ASD [54]. In ASD, prevalence rates or regression range between 32.1% [58] and 30% [59], with a mean onset age of 19.8 months [59]. In ASD, there are two patterns of regression: (I) a pattern of gradual regression during the first two years with seemingly typical socio-communicative development and (II) a pattern of socio-communication impairments before regression, which contrasts with CDD, which has typical development in the first 2 years of life [54]. However, CDD has been added to the diagnostic category of ASD in both DSM-V and ICD-11 and is considered a subtype of ASD. Diagnostic challenges regarding CDD and autistic regression have been raised previously [54]. Although questions remain on the diagnostic validity of CDD, the evidence does point towards distinct characteristics between CDD and ASD [60,61,62]. Evidence of DSRD is limited. The main criteria are the sudden onset of new neurological, psychiatric or mixed symptoms during a period of <12 weeks in an otherwise healthy person with Down syndrome and between 3–6 symptom clusters across eight clinical domains [51]. It often manifests in late adolescence [63], and the majority (90%) of patients with DSRD have language regression, followed by psychiatric symptoms such as mood dysregulation (42%), social withdrawal (34%) and anxiety (16%). Caregivers looking after those with DSRD also experience a higher burden when compared to those caring for individuals with Down syndrome [44]. Females are more affected [63], and features of catatonia and sleep disorders [64] have also been described in DSRD. In LKS, the onset ranges from 3 to 7 years [50], while in CDKL5-deficiency disorder (OMIM #300203) and FOXG1 disorder (OMIM #164874), the median onset of regression was 6.5 and 12 months, respectively [40]. Unlike RTT, in CDKL5-deficiency disorder, developmental delay is present from birth, and the seizures begin far earlier (<3 months) [65]. In *MECP2* duplication syndrome (MDS), regression has a later onset when compared to male RTT-like individuals [41], and regression in MDS affects about 40% of individuals [66].

Regression can be exacerbated by different factors (Figure 3), such as epilepsy. In MDS, regression was observed in 39% (12/31) of patients following seizure onset, of which 11 had treatment-resistant epilepsy [67]. Another study in MDS showed that regression was associated with the onset of epilepsy in 91% of patients but only 20% in those with spontaneous regression without epilepsy [68]. In those with generalised seizures, 69% had developmental regression in PHMDS in comparison to 33% that did not report generalised seizures [48]. Some other evidence has shed light on typical symptoms that may manifest before the onset of developmental regression [50], for example, loss of bladder or bowel function in CDD [50,54] or hypotonia and feeding difficulties in PHMDS [50]. In metachromatic leukodystrophy, in a subset of children, early developmental delay may also precede regression [45]. Table 1 summarises different aspects of regression. In the next section, we describe how *MECP2* interacts with the epigenome and how its dysfunction affects the neural epigenome, leading to neurological regression.

## 4. What Causes Neurological Regression in RTT?

### 4.1. Impact of MECP2 Across the Genomic Landscape

De novo pathogenic/likely pathogenic changes in the *MECP*2 gene within the long (q) arm of the X chromosome (Xq28) are the primary cause in 95% of cases with typical (classical) RTT. It has a global prevalence of 5–10 cases per 100,000 females [69], and RTT predominantly affects females; however, males can also have *MECP2* pathogenic changes, but the incidence is rare. Males with *MECP2* pathogenic/likely pathogenic variants usually die within the first two years of birth [29], and those that do survive post-birth have a range of clinical phenotypes [30,70]. There are several ways that *MECP2* may impact the genomic landscape. It can interact with histone deacetylase complexes [18], regulate RNA splicing [71], and assist RNA polymerase II to start transcription [72]. Two further mechanisms have been proposed that can shed light on the molecular pathogenesis of RTT. First, MeCP2 has condensate-partitioning properties that influence the separation of heterochromatin and euchromatin [19]. Pathogenic changes in *MECP2* can disrupt MeCP2 condensate-partitioning properties, implicating this mechanism in the pathogenesis of RTT. Second, co-repressors such as NCoR and SMRT can modulate chromatin by providing a docking platform or scaffold for transcription factors [73], and this leads to another molecular mechanism because *MECP2* mutations can disrupt the link between NCoR/SMRT co-repressors and chromatin [74]. Longer genes (>100 kb) are suggested to have a higher level of gene-body methylation when compared to shorter genes, and it has been shown that long genes are associated with neuronal development [75,76]. Long genes encode essential neuronal proteins and are selectively expressed in the brain [77], and MeCP2 cell-specific repression is biased for long genes [78]. Furthermore, MeCP2 is highly expressed in neuronal cells (16 × 10^6^ molecules per nucleus) [79]. These findings confirm that MeCP2 is an essential reader of methylated DNA in long genes and positions itself as an important protein laying down critical epigenetic marks across the genomic landscape. The epigenetic balance can be severely disrupted because of *MECP2* pathogenic variants.

The imbalance of the epigenome due to pathogenic changes in the *MECP2* gene does not provide the complete clinical picture of RTT pathophysiology, because individuals can have a pathogenic/likely pathogenic *MECP2* variant but have no obvious clinical signs or symptoms of RTT [80]. A translational cohort study has also identified rare *MECP2* variants in girls with precocious puberty without neurodevelopment problems [81]. Moreover, *MECP2* pathogenic variants are only found in about 95% of those with classic RTT and about 75% with atypical RTT [30,82]. More than 80 genes have been identified, and the changes lead to similar clinical characteristics seen in RTT that can be considered RTT/RTT-like [32]. However, no consensus exists for gene profiling studies regarding biological functions and variants of unknown significance. These findings demonstrate that *MECP2* pathogenic/likely pathogenic variants only support a diagnosis and are not confirmatory [22,30]. The diagnosis of RTT is independent of genetic profiles [24,83]. Most individuals with RTT who have a clinical diagnosis do have an *MECP2* pathogenic variant, and evidence from the natural history study has shown that 98% of individuals with a clinical diagnosis of classic RTT and 86% with a clinical diagnosis of atypical RTT had a pathogenic *MECP2* variant [84]. While other genes may be involved in RTT aside from *MECP2*, the next section will focus on neurological regression due to a pathogenic *MECP2* variant.

### 4.2. Neurological Regression

Regression forms part of the core diagnostic criteria for typical and atypical RTT [24,83]. Even though the epigenetic mechanisms influenced by MeCP2 are likely to be complex [85], the net effect of epigenome disruption caused by *MECP2* pathogenic variants is impaired neuronal functioning due to changes in the function or expression of neuronal genes. Even subtle changes to MeCP2 within GABA-releasing neurons can result in an RTT phenotype [86]. As mentioned by others [30], a key target gene modified by MeCP2 is brain-derived neurotrophic factor (*BDNF*) [87,88] and insulin-like growth factor 1 (IGF-1) [89]. In humans, the levels of MeCP2 increase during embryonic and post-natal development, stabilising at 10 years [90] (Figure 4). This profile reflects changes in transcriptional regulation and coincides with modifications in neuronal chromatin, such as size and shape [91,92], and enables MeCP2 to establish neuronal circuits during early developmental epochs and then later fine-tune cortical plasticity [93]. There are four stages in RTT [25,94], with neurological regression occurring after 6–18 months of seemingly normal development. However, the four stages of the disorder should not be viewed on their own. A personalised approach to management is needed, given the emergence of different symptom profiles at different stages of the disorder [30]. In one study, general regression of acquired skills (96%) followed by loss of hand use (88%), motor regression (84%), speech regression (84%) followed by behaviour regression (68%) were the most frequent regression forms observed [33]. In X-linked disorders, because of random X-chromosome inactivation, half of the cells will have mutated protein, while the other half will have normal (wild-type) protein. While most cases of classic RTT have balanced patterns of X-chromosome inactivation in the brain [95], a skewed pattern has also been reported, whereby one X chromosome is more inactivated than the other [96].

### 4.3. Genotoxic Stress and Neurological Regression

The manifestation of neurological regression due to impairments in the RTT neural epigenome is complex. Evidence has revealed some critical molecular insights into its mechanism. When viewing the pattern of *MECP2* dysfunction in an animal model, at the molecular level, Mecp2 impairments in female mice show different patterns of stage-specific and transcriptional changes compared to males [43]. Disease progression of the disorder may, therefore, not be due to autonomous transcriptional changes in specific cells but a failure of normal *MECP2*-expressing cells to counteract the changes caused by cells expressing mutant *MECP2* [43]. The suggestion that patterns of transcriptional dysregulation in normal cells are not able to compensate for changes in mutant cells, leading to staggered regression, could explain why neurological regression is seemingly delayed in RTT. Evidence suggests that MeCP2 expression is coordinated with central nervous system maturation, whereby the spinal cord and brainstem show positive expression before other developmentally newer brain regions, such as the hippocampus and cerebral cortex [97]. Furthermore, DNA methylation signatures vary in different brain regions, implying that epigenetic stability could be region-specific [98]. While there could be regional specialisation in the brain due to different signatures of DNA methylation, the kinetics of this process are unclear. The temporal stages when the consequences of impaired MeCP2 activity begin to emerge in different brain regions are unknown.

A DNA methylation atlas of the mouse brain has revealed spatial methylation gradients throughout the mouse brain [99]. While caution is needed in extrapolating this methylation atlas to the human brain, it is probable that the accumulation of methylated marks on the RTT epigenomic landscape and the resulting cumulative genotoxic stress could cause neurons to enter a senescent state and lead to rapid neurological decline [100] (Figure 5). Different spatial methylated gradients across the brain could influence the degree of genotoxic stress. For example, the cerebellum could be particularly sensitive to these gradients. Hence, motor deficits such as hand stereotypies could be the first to arise when RTT symptoms emerge [101]. Different gradients will influence the brain as it matures and lead to the emergence of specific symptoms at different developmental epochs. Recent evidence suggests that a reduction in histone H3 acetylation at downregulated genes could be one of the first molecular changes in the RTT epigenome before neurological dysfunction [102]. Most likely, these methylated gradients operate across the lifespan, but the genomic impact of MeCP2 across the neural epigenome is too small postnatally, and hence, overt symptoms do not appear. An accumulation of deleterious events across the epigenome could result in the emergence of RTT pathology [34,103]. There are likely to be subtle changes occurring at very early post-natal stages [34,35], where temporal changes due to MeCP2 dysfunction across the neural epigenome have not yet been consolidated enough for apparent signs of clinical symptoms to emerge. Some recent evidence from heterozygous *Mecp2* rats has also shown that sensory changes due to cortical defects occur before regression [42]. When viewed together, the concept of neurological regression in RTT needs to be revisited. Aberrant methylation gradients could be influenced by other factors to either speed up or delay neurological regression. As seen in other disorders that share features with RTT, epilepsy and infections could be precipitating factors [66]. Given that there is evidence of adolescent-onset neurological regression in RTT [104,105], other unidentified factors can alter methylation gradients to delay the onset of neurological regression. This may also apply to some other cases of RTT variants characterised by early-onset seizures (Hanefeld variant) or early regression (Rolando variant) [24,30,106].

### 4.4. Therapeutic Approaches

Dysfunction in the epigenome results in various epigenetic diseases, ranging from cancers to brain disorders [107]. While attention has mostly focused on deciphering the complexity of different events in the neural epigenome, there has been little attention focused on how these changes might lead to the neurological regression typically observed in RTT. Spatial methylation patterns could provide clues to how neurological regression manifests in RTT; however, while progress has been made, there remain questions on the trajectory of regression and the variability in its presentation. Aminoglycosides such as gentamicin have shown some promise in rare genetic disorders, such as RTT as a readthrough therapy [108], but their success is limited due to long-term toxicity. While advances have been made to reduce the dose of gentamicin [109], other strategies can also be considered. Therapeutic approaches have been tried in RTT with varied results [110]. In 2023, trofinetide ((2S)-2-{[(2S)-1-(2-aminoacetyl)-2-methylpyrrolidine-2-carbonyl]amino}pentanedioic acid) [111] was the first treatment approved by the Food and Drug Administration (FDA) for RTT [112]. The neuroprotective effects of trofinetide are suggested to be mediated via the insulin-like growth factor 1 (IGF-1) pathway. Through this pathway, trofinetide is thought to decrease neuroinflammation and neurotoxicity [112]. A dysregulated redox imbalance has been suggested to have a role in developmental regression in RTT [38], and a neuroinflammatory aetiology has been proposed for DSRD [113]. It is, therefore, conceivable that trofinetide, in theory, may help to reduce the deleterious events caused by neuroinflammation and neurotoxicity. To reduce the impact of regression in RTT, a personalised treatment approach would be required using biomarker-specific and neuroprotectant strategies. Future research could be directed towards restoring the balance of the neural epigenome in RTT. Epidrugs that target enzymes involved in epigenetic regulation may theoretically mitigate the effects of aberrant epigenetic changes across the RTT neural epigenome. Alternatively, epi-editing approaches using CRISPR-dCas9 may be used to precisely correct epigenetic defects in the RTT neural epigenome. These strategies have been suggested for autism spectrum disorder [114] and theoretically could also be feasible in individuals with RTT.

## 5. Summary

Given the widespread differences in genes and symptoms, RTT remains a clinical diagnosis. Identifying the genes responsible can help support the clinical diagnosis. In most cases of RTT, the *MECP2* gene is implicated, and dissecting interactions between *MECP2* and DNA/chromatin has provided further understanding of the disorder at the molecular level. While disruptions in epigenetic marks through methylation-dependent and independent mechanisms have provided clues on how *MECP2* alters transcriptional kinetics, its multifaceted role across the epigenome underscores its importance as an epigenetic modifier. Identifying that MeCP2 forms a liquid-like heterochromatin condensate when associated with DNA was an important milestone in furthering our understanding of the disorder [19]. Our narrative review has synthesised information on how MeCP2 operates at the molecular level and how *MECP2* pathogenic/likely pathogenic variants can lead to neurological regression.

### Limitations

The narrative review aimed to provide broader perspectives into neurological regression with a focus on RTT and was restricted to one search database. However, the broad nature of the present review means that some relevant articles in the literature could have been omitted. While the current review did not strictly meet the more rigorous PRISMA reporting guidelines for systematic reviews, the narrative literature review flowchart adopted PRISMA guidance to make the search strategy as robust as possible. The search strategy followed a defined structure, which included information on search terms, eligibility criteria and secondary searching. In summary, the narrative review synthesised the literature on how adverse events caused by pathogenic/likely pathogenic variants in *MECP2* impact the epigenome and provided meaningful insights on how this imbalance can lead to neurological regression in RTT.

## 6. Concluding Remarks

Rett syndrome (RTT) is an epigenetic disorder mainly affecting females and is associated with multisystem morbidity. There is no cure, and regular management of RTT symptoms requires a multidisciplinary approach. The symptoms of RTT reside on a broad clinical spectrum. Some prominent clinical signs are hand stereotypies and neurological regression between 6 and 18 months of age, following seemingly normal development. Pathogenic variants in *MECP2* lead to changes in the function or expression of neuronal genes. Due to the protective effect of X-inactivation, each cell in RTT has a mutated and healthy MeCP2 protein, implying that each cell has its specific epigenetic profile that together makes up the RTT epigenome. When the evidence is viewed together, the following findings emerge from this narrative review:

Expression patterns of MeCP2 coincide with brain maturation, and *MECP2* pathogenic/likely pathogenic variants result in adverse epigenetic events that accumulate across the RTT neural epigenome. This accumulation of genotoxic stress depends on different spatial methylated gradients across the brain. Some brain regions could be more sensitive to the aberrant gradients. The cerebellum is particularly vulnerable, and motor deficits such as hand stereotypies are the first symptoms.These gradients operate across the disorder’s lifespan, but MeCP2’s genotoxic impact on the epigenome is too small postnatally for obvious clinical symptoms to appear. Nevertheless, deleterious changes in the neural epigenome have already started, and the concept of a seemingly typical neurodevelopmental period before 6 months in RTT should be reframed.Epilepsy and recurrent infections could lead to the onset of regression in RTT; however, the trajectory of regression in this population is variable. The epigenetic factors that modify the onset (very early or adolescent onset) of neurological regression in RTT remain to be identified.A phased treatment approach would be needed. Neuroprotectants could be used at the early stages of regression. During progression, treatments that target symptom reduction and maintenance of function can be used. Biomarker-specific treatment personalisation may be required during regression to reduce impact.

## Figures and Tables

**Figure 1 ijms-26-05361-f001:**
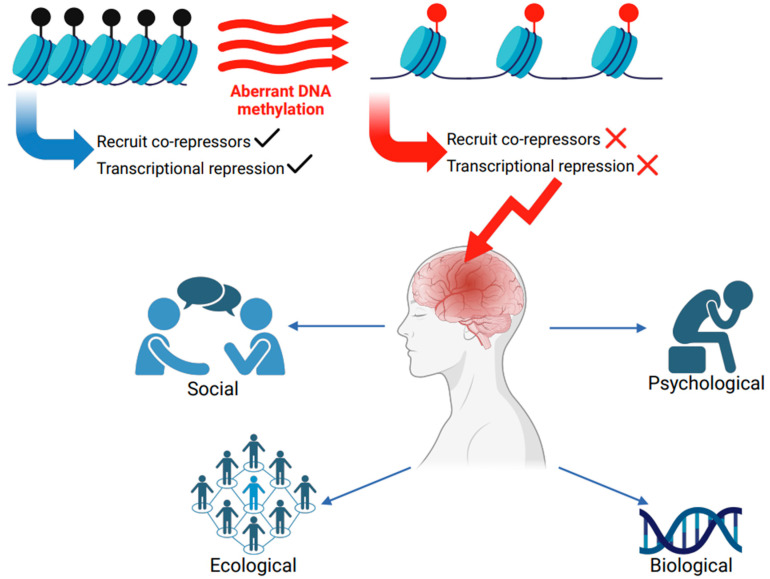
Epigenetics and onset of neurological disease. To maintain normal brain function, methylated DNA can influence DNA’s higher-order structure, i.e., chromatin packaging. When DNA methylation is not impaired, the kinetics of gene transcription can operate normally. However, disrupted DNA methylation alters the kinetics of gene transcription, changing the rates of different transcriptional mechanisms. A pathogenic/likely pathogenic change in the methyl-CpG-binding protein 2 (*MECP2*) gene results in a failure of MeCP2 to recruit co-repressors, resulting in an open chromatin configuration. For simplicity, the recruitment of co-repressors and transcription repression are shown. However, other kinetics of transcription could also be altered. Methylation impairment, in turn, leads to the manifestation of different neurological disorders. The impact of these disorders is far-reaching and has a significant biological (severity of the disease), ecological (burden on healthcare systems), psychological (mental health) and social (social support) impact. This figure was created using images from BioRender (https://biorender.com/).

**Figure 2 ijms-26-05361-f002:**
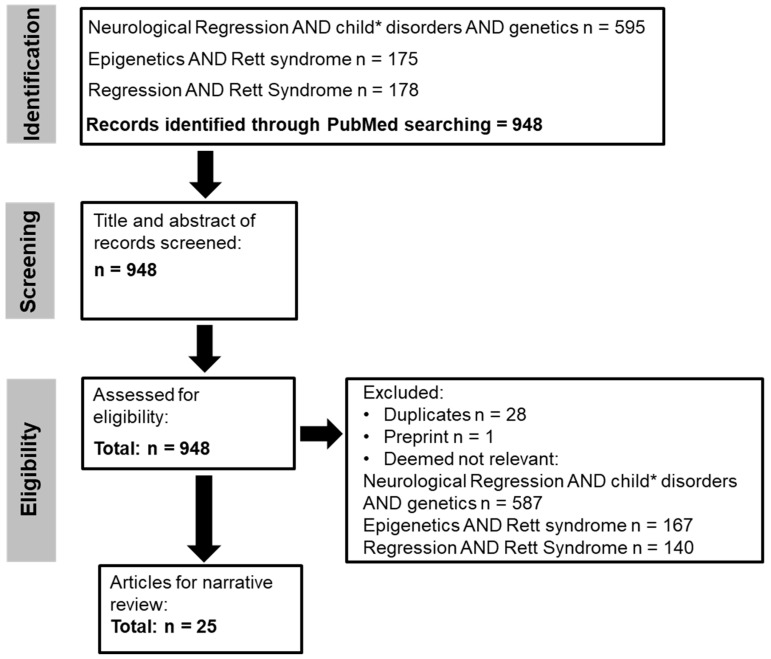
Narrative literature review flow chart.

**Figure 3 ijms-26-05361-f003:**
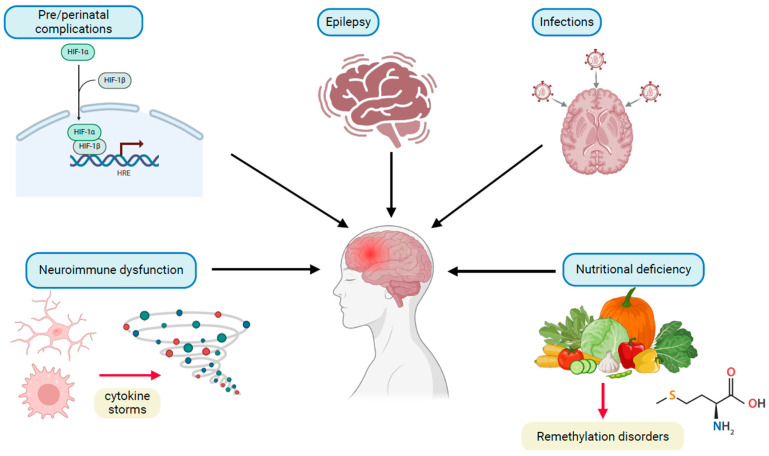
Contributing factors in neurological regression. Different factors can precipitate neurological regression. Pre/perinatal complications such as hypoxia at birth can activate pathways involving hypoxia-inducible factors (HIF 1 alpha and beta). Prolonged stimulation of these pathways can alter gene expression through hypoxia-responsive elements (HREs) and lead to neuronal cell death. The onset of regression may also be associated with epilepsy, as observed in some individuals with *MECP2* duplication syndrome. Infections of the brain can lead to changes in brain homeostasis. Infectious agents may lead to changes in brain development and longer-term neurological sequelae in the foetal brain. Neuroimmune dysfunction can lead to increases in pro-inflammatory cytokines, and the resulting cytokine storm can affect a variety of neuro-immune signalling cascades. These changes can lead to aberrant microglial activation, neurological deterioration and behavioural abnormalities. Deficiency in folate leads to remethylation disorders. Severe folate deficiency in childhood can lead to hyperhomocysteinemia, characterised by failure to thrive, developmental delay and epilepsy. This figure was created using images from BioRender (https://biorender.com/).

**Figure 4 ijms-26-05361-f004:**
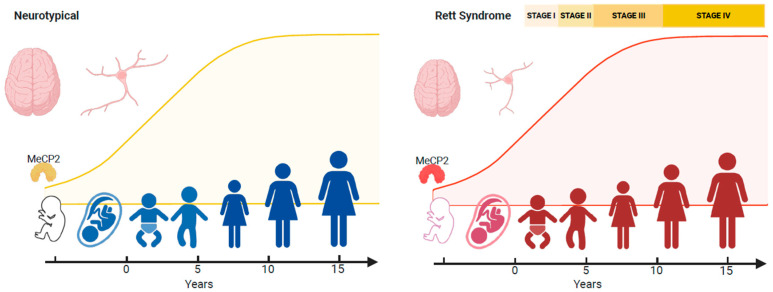
Expression levels of MeCP2. A diagram of MeCP2 expression levels across the lifespan. Expression levels of MeCP2 increase during the embryonic and post-natal period when neurogenesis and differentiation have been completed. Levels coincide with neuronal maturation and dendritic branching and stabilize at 10 years of age, at which point it maintains the functioning of mature neurons. In most cases of Rett syndrome (RTT), pathogenic/likely pathogenic variants in *MECP2* lead to volumetric reductions in brain size. Compared to neurotypical individuals, the effect size of volume reduction in RTT was the largest in the putamen, hippocampus and corpus callosum. Neuronal morphology is altered in RTT with less pronounced dendritic trees and fewer branches. This change leads to impairments in neuronal branching and synaptic connections. This figure was created using images from BioRender (https://biorender.com/).

**Figure 5 ijms-26-05361-f005:**
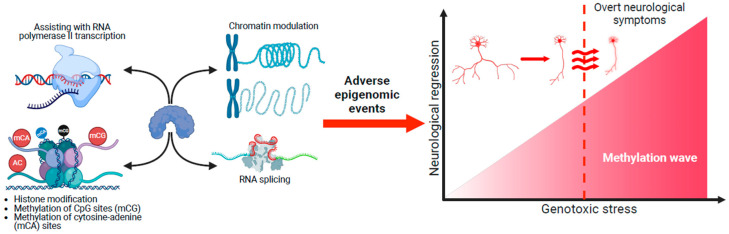
Adverse epigenomic events lead to neurological regression in Rett syndrome. The MeCP2 protein has varied roles across the genome. Pathogenic/likely pathogenic changes in *MECP2* can lead to widespread adverse epigenomic changes. Distinct parts of the brain could have region-specific methylation signatures and be more sensitive to these changes than others. Because MeCP2 expression coincides with brain maturation, the spinal cord and brainstem could be affected before other regions. Aberrant MeCP2 expression in the brainstem may, therefore, reflect the symptoms of autonomic dysregulation typically seen in RTT. The adverse impact of mutated MeCP2 across the epigenome is too little initially, as apparent symptoms do not appear. However, it is likely that during these initial stages, the neural epigenome is modified (for example, a decrease in histone H3 acetylation), leading to subtle prodromal changes. As adverse events accumulate across the epigenome, the build-up of genotoxic stress causes neurons to enter a senescent state, leading to neurological regression. Epigenetic stability is region-specific. The anecdotal cases of late-onset regression in RTT imply that other factors could be in play to counteract the accumulation of deleterious events across the epigenome and the emergence of early-onset regression typically seen in RTT. This figure was created using images from BioRender (https://biorender.com/).

**Table 1 ijms-26-05361-t001:** Key features of neurological regression in different disorders.

Condition	Comments	Reference
Age of Onset % Having Regression		
**Autism Spectrum Disorder (ASD)** 19.8 months. 32.1% to 30%.	Two patterns of regression (I) gradual regression during the first 2 years of life with typical socio-communicative development and (II) socio-communication deficits before regression. Regression is more gradual than observed in CDD.	[54,58,59]
**Childhood Disintegrative Disorder (CDD)** >24 months. Around 90% of individuals have some form of regression.	Loss of adaptive skills, i.e., bladder or bowel function appear before regression onset. Emerges gradually (6–12 months) but regression onset is quicker than seen in ASD.	[50,54,57]
**Down Syndrome Regression Disorder (DSRD)** Late adolescence. 90% have language regression.	Sudden onset over a period of less than 12 weeks. Catatonic symptoms and sleep disturbances frequently described. Females may be more affected.	[51,63,64]
**CDKL5-deficiency disorder** 6.5 months. ~37%.	Median age of seizure onset (~2 months) is quicker than other developmental encephalopathies. Developmental delay is from birth.	[40,65]
**FOXG1 disorder** 12 months. ~36%.	Median age of seizure onset is 11.5 months.	[40]
**Landa–Kleffner syndrome** ~3–7 years. Most will have regression.	Seizures may appear before regression onset.	[50]
**MECP2 Duplication syndrome** ~3–11.5 years (regression coincides with seizure onset/worsening). About 40% of individuals have regression.	Regression is more common in those with treatment-resistant epilepsy. Seizure onset is later than other developmental encephalopathies.	[41,67,68]
**Phelan–McDermid syndrome** About 6 years of age. Seen in 43% of individuals.	Hypotonia and feeding difficulties may manifest before regression onset. Seizures could be a risk factor for regression. Regression can appear in four stages: ⮚ Stage 1: Early-onset language regression (around 7 years). ⮚ Stage II: Plateau stage (can last up to 10 years). ⮚ Stage III: Neuropsychiatric decompensation with a mean onset of 20 years. ⮚ This is followed by Stage (IV): Late motor decline with an average onset age of 26 years.	[48,50,55,56]

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
