# Peer review of "Molecular Insights into Neurological Regression with a Focus on Rett Syndrome—A Narrative Review"

_ijms, 2025, doi:10.3390/ijms26115361_

Round 1
Reviewer 1 Report
Comments and Suggestions for Authors
The manuscript did a through and diligent review for current progress in MECP2 and Rett syndrome. It should be published after the following minor revision:
The authors claimed that, in line 260, "a skewed pattern has also been reported [76]." Please be more specific about what the skewness is - so the reader doesn't have to go back to the references and skim through the original paper for a simple fact.
Author Response
Please see my response to Reviewer 1 in the attached Word file.

Reviewer 2 Report
Comments and Suggestions for Authors
I had the pleasure of reviewing the manuscript by Singh et al., which explores how mutations in the MECP2 gene—central to Rett Syndrome (RTT)—lead to neurological regression through disruption of the neural epigenome. The authors emphasize that while prior research has largely focused on transcriptional mechanisms, less attention has been given to how epigenetic imbalances contribute to neuronal dysfunction and regression. Notably, they propose that methylation changes and genotoxic stress may drive neurons into a senescent state, and they outline the potential of epigenetic therapies and gene-editing approaches to restore epigenomic balance. This is a well-written and thoughtful manuscript, and I hope my suggestions help to further strengthen the work. Major Comments:
-
Manuscript Structure and Clarity:
Overall, the manuscript would benefit from additional editing to improve clarity and ensure the data are presented more objectively and systematically. -
Methodology Section:
I recommend including a flowchart in the methodology section to outline the review process—specifically, how many papers were identified, screened, included, and categorized under different themes. This would enhance transparency and reproducibility. -
Treatment Section:
I suggest adding a section discussing current or emerging therapeutic approaches, incorporating recent literature. For example, including references such as PMID: 39251501 could strengthen the discussion on potential interventions and increase the clinical relevance of the review. -
Minor Comments:
-
The term “mutation” is used throughout the text; I recommend replacing it with “pathogenic/likely pathogenic variants” to align with ACMG/AMP nomenclature.
-
Lines 81–116 appear misplaced—they read more like an introduction or discussion and should not follow the methods section.
-
Table 1 would be clearer if organized into three lines.
Author Response
Please see my responses to Reviewer 2 in the attached Word file.

Reviewer 3 Report
Comments and Suggestions for Authors
This is a well written paper reporting an overview about neurological regression related to gene methylation and MECP2 gene dysfunctions. The study is a literature review and offer a good landscape on these topics despite, in some instance, results a little simplistic. However the paper is readable and represent a good starting point for further in-deep analysis. Only few paper on the argument are available.
In the introduction a brief description of Rett sindrome should be inserted. Search strategy should be better explained. PRISMA diagram should be added. Manuscript structure is interesting. Table and figures are good. Bibliography is correct.
Author Response
Please see my responses to Reviewer 3 in the attached Word file.

Round 2
Reviewer 2 Report
Comments and Suggestions for Authors
Thank you for your reply to the beforementioned items. I am pleased with the modifications that the author's have made.
Author Response
Thank you for your comment.